# Expanding continual few-shot learning benchmarks to include recognition of specific instances

## Abstract

Continual learning and few-shot learning are important frontiers in progress towards broader Machine Learning (ML) capabilities. There is a growing body of work in both, but few works combining the two. One exception is the Continual Few-shot Learning (CFSL) framework of Antoniou et al. (2020). In this study, we extend CFSL in two ways that capture a broader range of challenges, important for intelligent agent behaviour in real-world conditions. First, we modify CFSL to make it more comparable to standard continual learning experiments, where usually a much larger number of classes are presented. Second, we introduce an 'instance test' which requires recognition of specific instances of classes – a capability of animal cognition that is usually neglected in ML. For an initial exploration of ML model performance under these conditions, we selected representative baseline models from the original CFSL work and added a model variant with replay. As expected, learning more classes is more difficult than the original CFSL experiments, and interestingly, the way in which image instances and classes are presented affects classification performance. Surprisingly, accuracy in the baseline instance test is comparable to other classification tasks, but poor given significant occlusion and noise. The use of replay for consolidation improves performance substantially for both types of tasks, but particularly the instance test.

## 1 Introduction

Over the past decade, Machine Learning (ML) has made enormous progress in many areas. The areas in which progress has been most dramatic share some common characteristics. Typically, a model learns from a large iid dataset with many samples per class and after a training phase, the weights are fixed i.e. it does not continue to learn. This is limiting for many applications, and as a result distinct subfields have emerged which embrace different learning requirements, such as continual learning and few-shot learning.

**Continual learning**   In continual learning (also known as lifelong learning), the challenge is to continually learn new tasks while maintaining performance on previous ones. A well known difficulty is catastrophic forgetting (McCloskey & Cohen, 1989) which recognises that new learning with different data statistics disrupts existing knowledge. There are many approaches to mitigate catastrophic forgetting that fall broadly into 3 categories (Delange et al., 2021): Regularization-based methods, which share representational responsibility between model parameters, Parameter isolation methods, which only modify a subset of parameters in response to new data, and Replay methods, which are inspired by Hippocampal replay (Parisi et al., 2019) and enable models to continue to learn slowly by internally repeating samples according to different policies.

**Few-shot learning**   In few-shot learning, only a few samples of each class are available. Few-shot learning is only rarely studied in combination with continual learning. In the standard framework (Lake et al., 2015; Vinyals et al., 2017), background knowledge is first acquired in a pre-training phase with many classes. Then one or a few examples of a novel class are presented for learning, and the task is to identify this class in a test set (typically 5 or 20 samples of different classes). Knowledge of novel classes is not permanently integrated into the network, which precludes the continual learning setting.

**Specific instances**  A special case of few-shot learning is reasoning about specific instances, which might have no class variance but are subject to variation caused by changing observational conditions, such as object or sensor pose, occlusion, and measurement noise. Reasoning about specific instances underpins memory for singular facts and an individual's own autobiographical history Baddeley (2001), and is therefore important for future decision making and planning.

Instance recognition is routine for humans and animals, but neglected in ML research. For example you usually know which coffee cup is yours, even if it appears similar to the cup of tea that belongs to your colleague. It is easy to see how this capability has applications across domains from autonomous robotics, to dialogue with humans or applications such as fraud detection. It is likely that this capability is accomplished through a combination DiCarlo et al. (2012) of object tracking Alvarez & Franconeri (2007), and recognition (or re-identification) by appearance. van Dyck et al. (2021) explores parallels between ML and human visual recognition of specific objects.

**Continual few-shot learning**  Another enviable characteristic of human and animal learning is the ability to perform both continual and few-shot learning simultaneously (CFSL). We need to accumulate knowledge quickly and may only ever receive a few examples to learn from. For example, given knowledge of vehicles (e.g. trucks, cars, bikes etc.), we can learn about any number of additional novel vehicles (e.g. motorbike, then skateboard) from only a few examples. CFSL, like the ability to recognise specific instances, is critical for everyday life, particularly artificial agents in dynamic environments and many industry applications.

**Establishing benchmarks in continual and few-shot learning**  While there are a number of established benchmarks in Continual learning and Few-shot learning individually, consensus regarding appropriate Continual *and* Few-shot learning benchmarks is still emerging Antoniou et al. (2020). Defining benchmarks is crucial for effective research progress, because benchmark conditions and characteristics strongly affect the potential performance of various models and algorithms. Frustratingly, many methods are only applied to one benchmark contender, making results incomparable to algorithms or models applied to alternative benchmarks.

**Instance test**  In this paper we seek to broaden the range of capabilities covered in CFSL benchmarks to include recognition of specific instances of an object, regardless of class. This is not equivalent to one-shot learning of classes.

The standard definition of 'classification' implies generalization - to learn to identify the class of an object, while ignoring individual variation. In contrast, learning specific instances can be conceived as a classification task without generalization, in which each exemplar is considered to be a class in its own right and individual variation is crucial for distinguishing these instances.

(Triantafillou et al., 2020) noted that learning coarse and fine classes is challenging in different ways. Similarly, learning specific instances requires an ability to learn the distinct characteristics of a specific instance of a class, to differentiate between very similar samples, and to differentiate samples of the same class.

The latter implies memorization, but simplistic memorization strategies will not be invariant to observational changes such as measurement noise, pose, or occlusion.

Learning instances has not been explored in a CFSL setting, but related challenges involved have been explored in the Few-Shot Instance Segmentation (FSIS) literature. Segmentation requires identification of all pixels in images or video belonging to an object, despite changes in pose, viewpoint, occlusions or illumination. FSIS aims to segment an object over multiple observations. A related task, Few-shot object detection (FSOD) concerns learning slowly to detect and segment all instances of specific classes Ganea et al. (2021). Creating segmentation training data is labour intensive, and impractical in many applications. This creates the need for few-shot segmentation; dynamically changing the set of relevant classes creates the *incremental* few-shot segmentation task, similar to the continual learning setting. Michaelis et al. (2019) describe one-shot instance segmentation on the MS-COCO dataset, which is unlabelled and includes realistic, high-resolution images.

This work approaches instance learning from the main body of CFSL benchmarks which utilise lower-resolution, object centered imagery. To develop multiple observations of an instance, we artificially introduce distortions such as noise and occlusion. While the task is not as graphically impressive as the FSIS work, it does allow evaluation of CFSL algorithms and models in the instance learning setting.

## 1.1 Contribution

This work contributes by: a) enhancing the CFSL framework, making it comparable to other continual learning benchmarks, and providing preliminary results for comparison; b) highlighting the relevance of instance learning, and increasing the scope of the CFSL benchmark by including an instance test; and c) exploring the potential contribution of replay on CFSL tasks.

# 2 Related work

## 2.1 Few-shot learning

MAML (Model-Agnostic Meta-Learning) Finn et al. (2017) is an early and influential Meta-Learning algorithm. Meta-learning is often described as "learning to learn", i.e. training to quickly acquire knowledge and exploit new data, while representing existing data in ways that generalise to new tasks. The paper differentiates fast acquisition of new tasks (few-shot learning), and the capability to leverage learning from previous tasks while learning new ones (meta-learning) as a solution to few-shot learning. The MAML paper also defined a popular few-shot learning benchmark including reinforcement learning, regression and classification tasks, the latter using Omniglot Lake et al. (2015) and Mini-ImageNet Vinyals et al. (2016).

These classification tasks have served as a standardized set of tasks or datasets on which various few-shot and meta-learning algorithms can be evaluated and compared. However, more recent work has argued for increasing the difficulty and complexity of tasks while continuing to introduce more capable algorithms to match.

Triantafillou et al. (2020) describe Omniglot and Mini-ImageNet Vinyals et al. (2016) as some of the most established benchmarks in few-shot learning but consider them too homogenous, limited to within-dataset generalization, and ignorant of relationships between classes; for example, they note that coarse-classification (e.g. dogs vs chairs) may be much easier than fine-classification (e.g. dog breeds). This is a theme our instance test, described below, takes even further.

In response, Triantafillou et al. created a Meta-Dataset for few-shot learning. The Meta-Dataset is larger – assembled from 10 pre-existing datasets, both episodic and non-episodic. In an episodic few-shot dataset, the data is organized into episodes. Each episode includes a support set and a query set. The support set contains a small number of labeled examples for each class or concept, while the query set contains unlabeled examples that need to be classified or predicted. A non-episodic dataset lacks the episodic structure, and is only divided into training, validation, and testing sets, without specific support and query sets.

Two of the ten datasets have a class hierarchy, enabling coarse-class and fine-class tasks. The Meta-Dataset benchmark is parameterised in terms of Ways (the number of classes) and Shots (the number of training examples). They explore the comparative performance of pre-training and meta-learning using several models, including Meta-learners, Prototypical Networks, Matching Networks, Relation Networks, and MAML. Their contribution - in addition to the Meta-Dataset – is Proto-MAML, a meta learner that combines Prototypical Networks and MAML.

## 2.2 Continual few-shot learning

Antoniou et al. (2020) define their Continual few-shot learning (CFSL) benchmark as a series of few-shot learning tasks. In addition to the existing Omniglot dataset, they propose the SlimageNet64 dataset for this purpose, a "slim" version of the ImageNet dataset with only 200 instances of each class at low resolution (64x64 pixels).

Antoniou et al. (2020) classify approaches to few-shot learning as: 1) Embedding or Metric learning; 2) Gradient based meta-learning, such as MAML; 3) Hallucination based - using the aforementioned methods or another generative process to augment the support set of known exemplars; 4) Other methods, including Bayesian approaches such as relational networks Santoro et al. (2017).

They compare a set of popular few-shot algorithms on this dataset, including methods that pretrain a model and then fine-tune and meta-learning approaches. Their work is the basis for the enhanced benchmark proposed in this paper.

Caccia et al. (2020) propose a CFSL benchmark called OSAKA (Online Fast Adaptation and Knowledge Accumulation). OSAKA requires models to learn new, out-of-distribution tasks as quickly as possible, while also remembering older tasks. Out of distribution shifts include replacing one dataset (such as Omniglot) with another (such as MNIST). OSAKA deliberately blurs the boundaries of episodes, and focuses on the tasks currently being evaluated instead. Tasks can re-occur, and new tasks appear. Performance is measured cumulatively, during new class introduction, not only finally after exposure has completed. The shifts between tasks are stochastic, and are not observable to the model (the authors note that some continual learning methods such as EWC rely on this knowledge). The target distribution is a context-dependent, non-stationary problem.

Caccia et al. (2020) envisage that OSAKA is a very challenging benchmark. They provide a number of reference models, several based on MAML, and propose Continual-MAML, which detects and react to out of distribution data. Most methods perform badly under OSAKA conditions.

### 2.3   Beyond episodic continual learning

Time-series data also motivates a move away from discrete, detectable episodes. Harrison et al. (2020) sought to eliminate known and abrupt task transitions or episode segmentation. They look at time-series data, where latent task variables undergo discrete, unobservable, stochastic changes. Observable data is dependent on the latent task variables. For time-series CFSL they propose Meta-learning via Changepoint Analysis (MOCA) - a meta-learning algorithm with a changepoint detection scheme. Their benchmark has two phases - meta-learner training, and online adaptation (evaluation).

Ren et al. (2021) also extend few-shot learning to an online, continual and *contextual* setting, with online evaluation while learning novel classes, like OSAKA. Contextual here refers to a changing, partially-observable process which affects the desired classification task. Context information is provided as a background to image classification datasets, including Omniglot and ImageNet. The contextual task tests an agent's ability to quickly learn the meaning of a class in a specific context, thereby adapting to that change.

### 2.4   Replay methods for continual learning

A variety of replay methods have been used for continual learning, surveyed by Parisi et al. (2019); Bagus & Gepperth (2021). Replay methods are largely inspired by Complementary Learning Systems (CLS) (McClelland et al., 1995; O'Reilly et al., 2014; Kumaran et al., 2016), a computational framework for learning in mammals. In CLS, a short-term memory (STM) representing the hippocampus stores representations of specific stimuli in a highly non-interfering manner. Interleaved replay to a long-term memory (LTM) enables improved learning and retention. The LTM is often assumed to be a model of the neocortex, and is considered to be an iterative statistical learner for structured knowledge.

The most common replay approach is to store samples in a buffer, and replay them during training. One of the challenges is memory capacity, leading to various strategies for selecting the subset of training data to store. Strategies include maximising novelty Gepperth & Karaoguz (2016), using lower dimensional latent representations Pellegrini et al. (2020); van de Ven et al. (2020), maximising diversity Aljundi et al. (2019), and selecting for an equal distribution over classes, as in NSR+ in Bagus & Gepperth (2021). There are also different strategies for using the buffer contents in new learning, broadly grouped into Rehearsal and Constraint methods. In rehearsal, buffer samples (typically randomly sampled) are presented for training, either in sleep phases Gepperth & Karaoguz (2016), or interleaved with new training data. Constraint methods constrain learning of new tasks using buffered samples e.g. so that the loss does not increase with

new tasks Lopez-Paz & Ranzato (2017). Another approach to replay, inspired by the generative nature of the hippocampus, is to generate representative samples from a probabilistic model and interleave with new tasks Shin et al. (2017); Maracani et al. (2021); Stoianov et al. (2022). These approaches have dramatically reduced memory requirements.

## 3 Experimental method

We first give an overview of the CFSL framework (Antoniou et al., 2020), upon which our study is based, in Section 3.1. Second, we describe experiments to scale selected tests from (Antoniou et al., 2020), referred to as 'CFSL at scale', in Section 3.2. Third, we introduce the instance test in Section 3.3. Then, we describe the training process and models used in Section 3.4 and finally describe the experimental setup, Section 3.5. The source code for all experiments is located at `https://github.com/xxxx` (NOTE: the URL is not given to protect anonymity).

### 3.1 Continual few-shot learning framework – background

As context, we will recap the relevant method and terminology used in the CFSL framework. In continual learning, new tasks are introduced in a stream and old training samples are never shown again. Performance is continually assessed on new and old tasks. In the CFSL framework, the new data is presented with groups of samples defined as 'support sets', and then the model must classify a set of test samples in a 'target set'. The target set contains samples from all of the classes shown in that episode. The experiment is parameterized by a small set of parameters described in Table 1. By varying these parameters, the experimenter can flexibly control the few-shot continual learning tasks – total number of classes (NC), samples per class, and the manner in which they are presented to the learner. A visual representation is shown in Figure 1.

Table 1: The parameters that fully define an experiment in the CFSL framework by Antoniou et al. (2020)

| Parameter | Description |
|---|---|
| NSS | Number of support sets |
| CCI | Class-change interval e.g. if CCI=2, then the class will change every 2 support sets |
| $n$-way | Number classes per set |
| $k$-shot | Number of samples per support class in a support set |

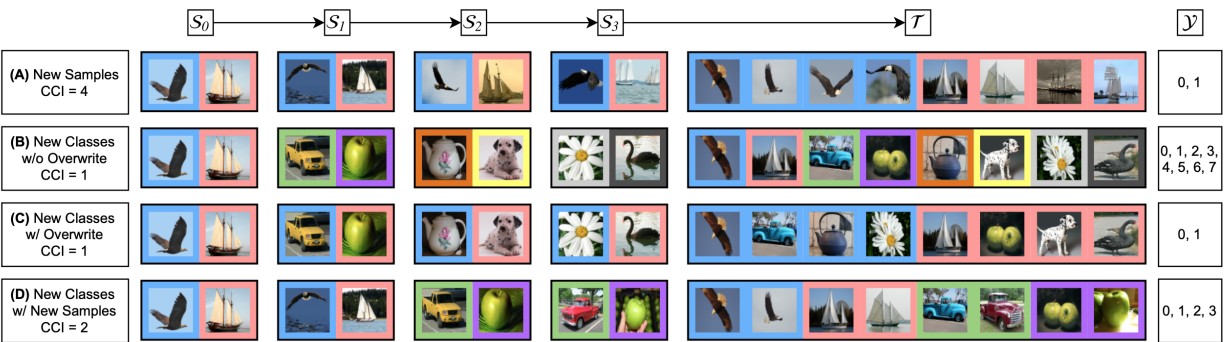

Visual representation of the four continual few-shot task types. Each row corresponds to a task with Number of Support Sets, NSS=4, and a defined Class-Change Cnterval (CCI). Given a sequence of support sets, $\mathcal{S}_n$, the aim is to correctly classify samples in the target set, $\mathcal{T}$. Colored frames correspond to the associated support set labels.

Figure 1: **Visual representation of CFSL experiment parameterization.** Reproduced from (Antoniou et al., 2020).

### 3.2 CFSL at scale

In the first set of experiments, we replicated a representative set from Antoniou & Storkey (2019), but extended the number of classes. The objective was to provide results that are more comparable to other continual learning studies in the field.

We chose to base our experiments on the parameters of Task D (see Figure 1) as described in the original CFSL benchmark (Antoniou et al., 2020) as it resembles the most common, applicable real-world scenario – it introduces both new classes and additional instances of each class.

#### 3.2.1 Framework baseline – replicating the original experiments

During our work with CFSL, we identified and fixed a number of issues in the original CFSL codebase (Antoniou et al., 2020), and collaborated with the authors to have them reviewed and merged upstream. Given the significance of some of these issues, we opted to replicate a selected number of the original experiments to properly contextualise our new experiments and results. The main issues related to a) Pretrain+Tune weight updates and b) mislabelling of new instances which became an issue where CCI>1.

#### 3.2.2 Scaling

It is common in the continual learning field for the number of classes to range from 20 to 200, even if the number of tasks in a sequence may be small (approximately 10). Therefore, we introduced experiments with up to 200 classes (compared to 5 classes per support set and a maximum of 10 support sets in (Antoniou & Storkey, 2019)). We experimented with presenting the classes in two ways: **Wide**, in which the number of support sets was small but with a larger number of classes per set, and **Deep**, where there were a larger number of support sets but with a smaller number of classes per set. See Figure 2.

In a real-world setting, the way that the samples are presented, Wide vs Deep, is not directly tied to how the learner experiences new classes, but rather a choice about training method. For example, the same stream of samples could be organized as Wide or Deep. However, Wide may be more suited to scenarios where you are exposed to a wider range of new classes simultaneously, like exploring a completely new environment, whereas Deep may be better suited to learning incrementally about a narrow environment or task.

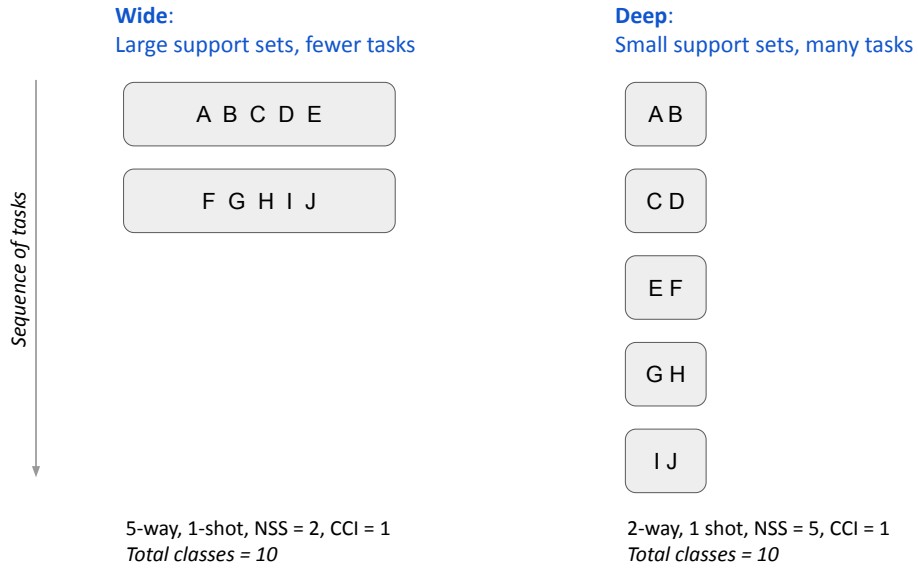

Figure 2: **Wide vs Deep.** An illustration of Wide vs Deep experiments. Wide have big support sets and few tasks, Deep has small support sets and many tasks.

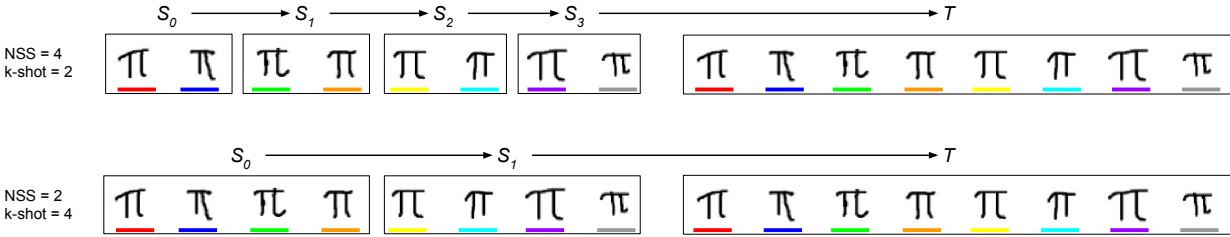

Figure 3: **Instance test:** Two configurations with 8 instances. In the first configuration, there are 4 support sets $(S_n)$, in the second there are 2. The test set (T) consists of the same instances that were shown throughout the support sets. For each test sample, the challenge is to identify the matching identical instance from the support sets, amongst other highly similar instances. NSS = Number of support sets, and k-shot = the number of instances that are shown for a given class. The samples are color coded to show which ones are identical.

Two of the replication configurations were used as baselines, with 10 and 20 classes. Then, we created Wide and Deep configurations, with 20 total classes like the 2nd baseline, but we modified the way that the classes were presented. Finally, the number of classes was increased ten-fold to 200, presented in both Wide and Deep configurations. In all of the experiments, $k$-shot is set to 1, so for any support set, there is only one exemplar per class.

### 3.3 Instance test

As discussed above, we seek to increase the range of natural conditions and challenges that are captured by the CFSL benchmark, to include recognition of a specific single object - one instance of a class. We call this an 'instance test'. The test is based on the instance test in AHA (Kowadlo et al., 2020). In the instance test, the learner must learn to recognize specific exemplars amongst sets where all the exemplars are drawn from the same class.

Like in the regular tests, an episode consists of several support sets, see Figure 1. However, in the instance test, all samples are drawn from the same class and are therefore very similar to each other. The test set consists of the same instances, and the challenge is classify each instance i.e. which of the support set instances corresponds to a given identical instance in the test set. It is a challenging problem because they are very similar to each other. A helpful way to think of it is classification of similar classes, and each class has only 1 sample. The test is illustrated in Figure 3.

Due to the flexibility of the CFSL framework, the instance test can be implemented as a special case of the existing parameters. $n$-way is set to 1, so that there is only one class in each support set. CCI is equal to NSS so that there is no class change between support sets. Then the $k$-shot or samples per class, determines how many instances are shown for a given class, which we refer to as Number of Instances, NI. We used a constant total number of instances for all experiments, 20, but experimented with presenting the samples differently, in terms of number and size of support sets. We reused empirically optimal hyperparameters from the Scaling Experiments.

A feature of the instance tests is the addition of increasing levels of corruption in the form of noise and occlusion. Even specific instances are subject to these forms of variation. We used higher levels for SlimageNet64 in order to achieve the same deterioration of accuracy. Note that SlimageNet64 images have background content which can be used for recognition even given substantial noise and occlusion.

### 3.4 Training methods and architectures

We selected three baseline methods from the original CFSL paper Antoniou et al. (2020) to represent each family of algorithm that was tested, using the implementations published in their open-source code-

base (`https://github.com/AntreasAntoniou/FewShotContinualLearning`). The first method was Pre-train+Tune using a CNN architecture based on stacking convolutional VGG blocks Simonyan & Zisserman (2014). The second method was Prototypical Networks (ProtoNets) (Snell et al., 2017), using the same underlying network architecture. We intended to also evaluate SCA Antoniou & Storkey (2019), which is a complex and high performing meta-learning approach. However, we encountered resource constraints and were unable to successfully complete the larger variant of each experiment type.

To ensure a fair comparison between models and experiments with varying number of tasks/classes, we optimized hyperparmeters for each experiment, using univariate sweeps. The hyperparameter search included the architecture space (number of filters, number of VGG blocks and learning rate). Under these conditions, both Protonets and Pretrain+Tune methods could explore the same architectures.

Experiments were run on Omniglot Lake et al. (2015) and SlimageNet64 Antoniou et al. (2020) datasets. Each was split into training, validation and test sets. The SlimageNet64 splits were chosen to ensure substantial domain-shift between training and evaluation distributions to provide a strong test of generalization Antoniou et al. (2020).

All the models were pretrained until plateau (30-50 epochs for SlimageNet64 and 10 epochs for Omniglot) with 500 update steps per epoch. At the end of each epoch, the models were validated on the CFSL tasks (200 episodes consisting of support and target test sets as described above in Section 3.1). At the conclusion of pre-training, the best performing model was selected and tested on the CFSL tasks. Experiments were repeated 5 times with a different random seed for data sampling and model initialization. For all the models, weights were initialized with Xavier, uniform random distribution. For pre-training, the Adam optimizer was used with cosine annealing scheduler and weight decay regularization (value=0.0001). For fine-tuning, the optimizer was a straightforward gradient descent optimizer without momentum.

For comparison with Antoniou et al. (2020) in the 'framework baseline' replication experiments (Section 3.2.1), we used a 5-model ensemble. For the other experiments, we preferred to see a more direct measure of performance and used a single model.

### 3.4.1 Pretrain+Tune

In this method, the model is trained on a large corpus prior to the CFSL tasks. Then during the CFSL tasks, the model is fine-tuned on the support set images before being evaluated.

The architecture consists of a VGG-based architecture with a variable number of blocks and a head with 1 dense linear layer. Like in VGG, the blocks consisted of a 2d convolutional layer with a variable number of 3x3 filters with stride and padding of 1, leaky ReLU activation function, a batch norm layer and a max-pooling layer with stride of 2 and a 2x2 kernel. See Appendix 7.1 for more details.

A distinct classification head was used for the pretraining phase, with the same number of classes as the pretraining set. At the end of pre-training, the top $q$ models were chosen using the validation set. $q$=5 for the ensembling approach used in the Replication experiments, and 1 for all the others. Then in evaluation, for each task, all the weights were reset to the pretrained values and a newly initialized head was used. Fine-tuning adjusted the weights throughout the network including the VGG blocks as well as the classifier head. The process is illustrated in Figure 4.

### 3.4.2 ProtoNets

Prototypical Networks (ProtoNets) train a network to learn an embedding that is optimized for matching Snell et al. (2017). The embedding for each class is considered to be a prototype for that class. In these experiments, the architecture details and hyperparameterizations are very similar to Pretrain+Tune above, except that no bias was used and no dense layer was required (as the learning objective is calculated from the embedding without the need for a standard classifier).

The same dataset splits were used as for Pretrain+Tune. In this case the pretraining split was used to learn the embedding, and the evaluation split was used for the tasks in the same way. Cosine similarity was used

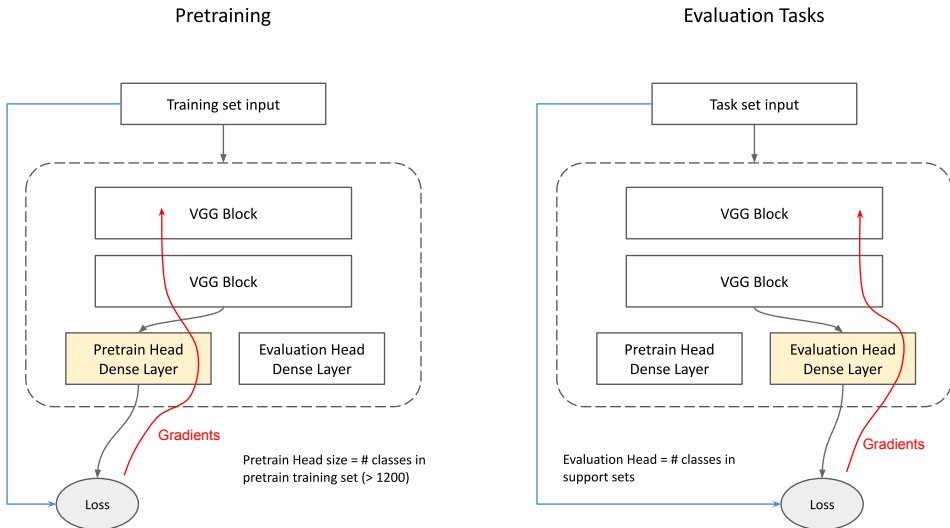

Figure 4: **Pretrain+Tune Training and Evaluation:** Different heads are used for pretraining and each task, because the number of classes varies between pretraining and task settings. All trainable parameters in the head and VGG blocks are adapted during pretraining and fine-tuning.

to calculate the distance between embeddings to perform classification. No learning occurred during the evaluation tasks.

### 3.4.3   Learning with replay

As described in the related work (Section 2.4), replay methods have been applied to continual learning Parisi et al. (2019); Bagus & Gepperth (2021), but not to continual few-shot learning. In this work, we created a very simple replay mechanism to provide initial exploration of the performance benefit of replay methods. It was applied to the Pretrain+Tune method, and is referred to as Pretrain+Tune+Replay.

The Pretrain+Tune+Replay architecture is shown in Figure 5. The long-term memory (VGG network) is augmented with a short-term memory (STM), consisting of a circular buffer, in which new memories replace older memories in a FIFO (first in, first out) manner. The STM stores samples from recent tasks and replays them by interleaving samples from the STM with samples from current tasks during fine-tuning. The process is divided into two stages. First, the current support set is stored in the STM, adding to recent support sets. $b$ is the buffer size, measured in support sets, and is a tuneable hyperparameter. Second, the network is trained using the current support set, as well as samples randomly drawn from the replay buffer. The number of samples is determined by a second hyperparameter $p$. The hyperparameter search was expanded to include both $b$ and $p$ in the replay experiments.

In this initial exploration of replay for CFSL, we applied it only to the Pretrain+Tune method. Replay fits naturally, as weights are already adapted during fine-tuning and it follows the precedent of other replay methods in the literature (described in related work, Section 2.4). In contrast, the conceptual approach of ProtoNets is to meta-learn a fixed embedding using a wide spread of classes from a background data set and generalize well from that, without further modifications. Replay for ProtoNets would be an interesting challenge for future work.

### 3.5   Experimental setup

The code was written using the PyTorch framework v1.6.0. The Omniglot experiments were conducted on two machines. Machine 1 had a GeForce GTX 1070 GPU with 8GB RAM and an Intel Core i7-7700 with 32GB RAM. Machine 2 had a GeForce GTX 1060 GPU with 6GB RAM and an Intel Core i7-7700 with

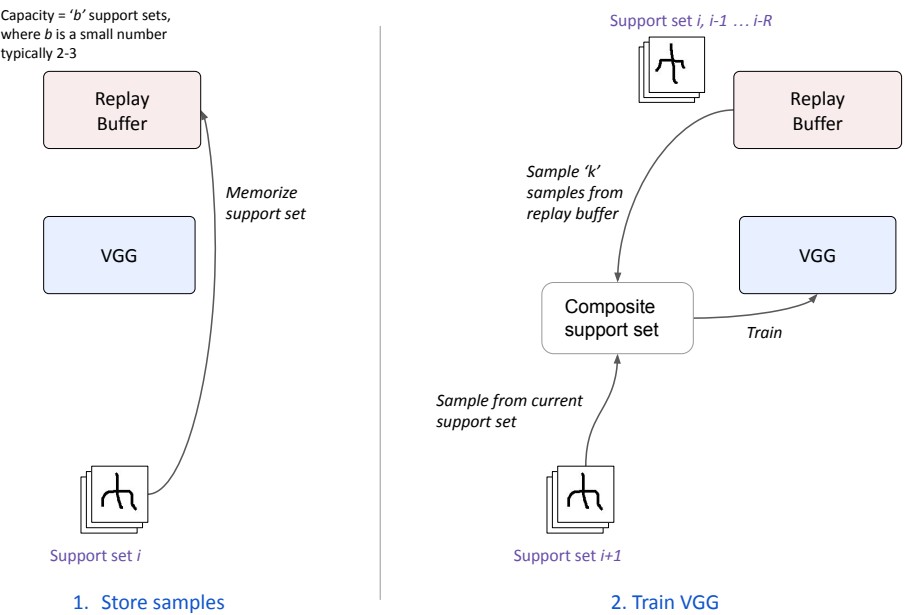

Figure 5: **Learning with replay.** Complementary Learning Systems (CLS) setup with Long Term Memory (LTM), paired with a circular buffer Short Term Memory (STM). First, in a memorization step, the STM temporarily stores recent support sets. Second, in a recall step, the memorized data are used in LTM training.

16GB RAM. The SlimageNet64 experiments were conducted on cloud compute using virtual machines with an A10 GPU with 24GB and 30 vCPUs, 200GB RAM, using Lambda Labs `https://lambdalabs.com/`.

The duration of training for an individual experiment was in the order of half a day, including pretraining and multiple seeds.

## 4 Results

### 4.1 CFSL at scale

#### 4.1.1 Framework baseline – replicating the original experiments

The results of the replication experiments are summarised in Table 2, which includes reference values from (Antoniou et al., 2020) for comparison. In the experiments that were affected by code fixes (CCI>1), performance improved substantially from unusually low values, and the performance across experiments followed a more expected trend (i.e. increasing accuracy of Pretrain+Tune with decreasing number of classes). ProtoNets was substantially more accurate than Pretrain+Tune, and performed consistently across different variations of the presentation of 10-50 total classes.

#### 4.1.2 Scaling – Omniglot

The results are summarised in Table 7 and Figure 6 (a). See Appendix 7.4 for details of hyperparameters used. The number of fine-tuning training steps for the replay experiments, which were reduced to allow it to run within our hardware RAM constraints, are shown in Appendix 7.2.

Overall, increasing the number of classes from 20 to 200 led to a dramatic decrease in accuracy. The manner in which the classes were presented ('Wide' or 'Deep') affects performance substantially. The Protonets method had the best accuracy. The Pretrain+Tune method was substantially improved by the addition of Replay, reaching a performance similar to Protonets in Baseline 1.

Table 2: **Replication experiments.** Replication of Task D from (Antoniou et al., 2020) after correcting errors in the framework code. Accuracy is shown in %, as mean ± standard deviation across 3 random seeds.

| Method name | NSS | CCI | $n$-way | $k$-shot | Number of classes | Ensemble accuracy | Accuracy | Reference ensemble accuracy |
|---|---|---|---|---|---|---|---|---|
| **Pretrain + Tune** | 4 | 2 | 5 | 2 | 10 | $37.81 \pm 0.77$ | $36.7 \pm 0.80$ | $7.91 \pm 0.15$ |
| **Pretrain + Tune** | 8 | 2 | 5 | 2 | 20 | $27.92 \pm 0.10$ | $26.41 \pm 0.14$ | $3.86 \pm 0.06$ |
| **Pretrain + Tune** | 3 | 1 | 5 | 2 | 15 | $17.76 \pm 0.32$ | $17.40 \pm 0.33$ | $9.97 \pm 0.14$ |
| **Pretrain + Tune** | 5 | 1 | 5 | 2 | 25 | $13.76 \pm 0.08$ | $13.10 \pm 0.03$ | $6.02 \pm 0.02$ |
| **Pretrain + Tune** | 10 | 1 | 5 | 2 | 50 | $9.73 \pm 0.06$ | $8.36 \pm 0.05$ | $3.13 \pm 0.03$ |
| **ProtoNets** | 4 | 2 | 5 | 2 | 10 | $97.93 \pm 0.05$ | $96.98 \pm 0.05$ | $48.98 \pm 0.03$ |
| **ProtoNets** | 8 | 2 | 5 | 2 | 20 | $96.66 \pm 0.03$ | $95.22 \pm 0.06$ | $48.44 \pm 0.03$ |
| **ProtoNets** | 3 | 1 | 5 | 2 | 15 | $97.12 \pm 0.06$ | $95.88 \pm 0.12$ | $95.30 \pm 0.12$ |
| **ProtoNets** | 5 | 1 | 5 | 2 | 25 | $95.93 \pm 0.12$ | $94.36 \pm 0.05$ | $91.52 \pm 0.20$ |
| **ProtoNets** | 10 | 1 | 5 | 2 | 50 | $92.43 \pm 0.27$ | $90.24 \pm 0.10$ | $83.72 \pm 0.19$ |

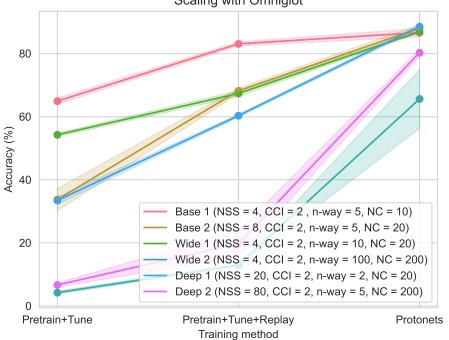

(a) Scaling experiments: Omniglot images

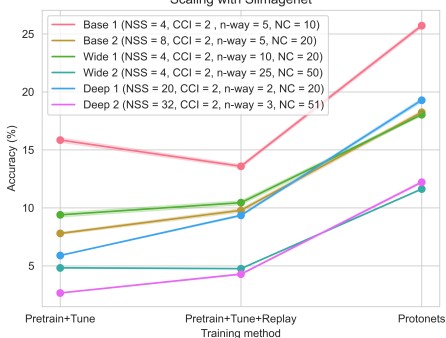

(b) Scaling experiments: SlimageNet64 images

Figure 6: **Scaling experiments.** In the CFSL scaling experiments, the number of classes was increased to 200. Three approaches were compared: i. Pretrained network with fine-tuning, ii. Pretrained network with fine-tuning and the addition of Replay, and iii. Protonets. The bold line shows the mean, and the shaded area shows one standard deviation, across 5 random seeds.

### 4.1.3 Scaling – SlimageNet64

Results for SlimageNet64 experiments are summarised in Table 8 and Figure 6 (b). See Appendix 7.4 for details of hyperparameters used. Overall, accuracy is lower than Omniglot images; Protonets delivered the best accuracy. The benefit of Replay was less dramatic, but still present in all settings except Baseline 1.

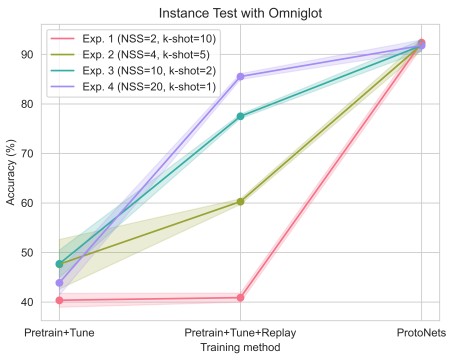
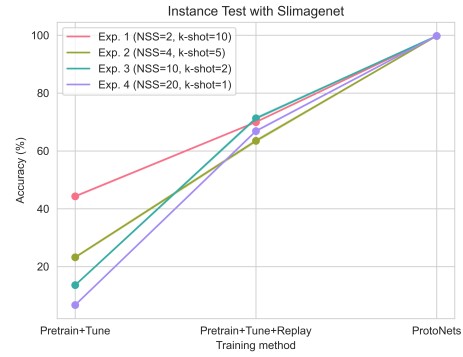

(a) Instance test: Omniglot images    (b) Instance test: SlimageNet64 images

Figure 7: **Instance test.** In the instance test, $k$-shot translates to the size of the support set. It is 1-shot in the sense that each instance is only shown once. NI, number of instances = 20 for all the experiments. The bold line shows the mean, and the shaded area shows one standard deviation, across 5 random seeds.

## 4.2 Instance test

The results for the instance test are summarised in Table 3 (Omniglot), Table 4 (SlimageNet64) and Figure 7. The number of 'items to identify', which in this case are separate instances, is constant at 20 for all of the experiments. Five experimental settings were tested, varying parameters NSS, $k$-shot and NI.

Model hyperparameters from Baseline 2 were reused due to the similarity of the setting, in lieu of further hyperparameter search. The number of fine-tuning training steps for the replay experiments, which had to be reduced to allow it to run within our hardware constraints, are given in Appendix 7.2.

Compared to classification experiments, accuracy is relatively high for Protonets. The Pretrain+Tune method is noticeably worse than Protonets, and as observed in classification experiments, Replay provides a substantial improvement in accuracy, but in most experiments does not provide accuracy comparable to Protonets.

Table 3: **Instance test: Omniglot.** Accuracy is shown in %, as mean ± standard deviation across 5 random seeds. $n$-way=1 for all experiments, to restrict distinguishing between similar instances of a single class. In the instance test, $k$-shot translates to the size of the support set. It is 1-shot in the sense that each instance is only shown once. NI, number of instances = 20 for all the experiments.

| Method name | Exp. 1 NSS=2, $k$-shot=10, NI=20 | Exp. 2 NSS=4, $k$-shot=5, NI=20 | Exp. 3 NSS=10, $k$-shot=2, NI=20 | Exp. 4 NSS=20, $k$-shot=1, NI=20 |
|---|---|---|---|---|
| Pretrain+Tune | $40.35 \pm 1.40$ | $47.63 \pm 5.00$ | $47.72 \pm 2.77$ | $43.88 \pm 2.47$ |
| Pretrain+Tune+ Replay | $96.39 \pm 1.36$ | $89.19 \pm 2.79$ | $82.77 \pm 2.15$ | $79.46 \pm 3.01$ |
| ProtoNets | $92.38 \pm 1.00$ | $91.94 \pm 1.09$ | $91.79 \pm 1.16$ | $91.79 \pm 1.16$ |

## 4.3 Instance test – with noise and occlusion

Figures 8 and 9 illustrate the results of the three methods in Instance Test Experiments 1-4 under varying levels of noise and occlusion. Details of experiment configurations can be found in Tables 3 and 4.

Table 4: **Instance test: SlimageNet64.** Accuracy is shown in %, as mean ± standard deviation across 5 random seeds. $n$-way=1 for all experiments, to restrict distinguishing between similar instances of a single class. In the instance test, $k$-shot translates to the size of the support set. It is 1-shot in the sense that each instance is only shown once. NI, number of instances = 20 for all the experiments.

| Method name | Exp. 1 NSS=2, $k$-shot=10, NI=20 | Exp. 2 NSS=4, $k$-shot=5, NI=20 | Exp. 3 NSS=10, $k$-shot=2, NI=20 | Exp. 4 NSS=20, $k$-shot=1, NI=20 |
|---|---|---|---|---|
| Pretrain+Tune | $44.32 \pm 0.27$ | $23.23 \pm 0.30$ | $13.62 \pm 0.18$ | $6.73 \pm 0.07$ |
| Pretrain+Tune+Replay | $69.98 \pm 0.18$ | $63.49 \pm 0.45$ | $71.29 \pm 0.44$ | $66.88 \pm 0.22$ |
| ProtoNets | $99.72 \pm 0.07$ | $99.77 \pm 0.05$ | $99.76 \pm 0.05$ | $99.78 \pm 0.06$ |

With smaller amounts of noise and occlusion, results and in particular the ranking of the three methods are unchanged. However, with larger amounts of noise and occlusion, the Pretrain+Tune+Replay method is often more accurate than Protonets on both image datasets.

## 5 Discussion

In this study we found that for the methods tested, few-shot continual learning of new classes is more difficult at scale i.e. as the number of classes was increased from 20 to 200.

Performance of all methods on the novel instance test was comparable to performance on similarly sized classification tasks at scale.

The ProtoNets method outperformed Pretrain+Tune method in all tasks. With the addition of replay, Pretrain+Tune accuracy improved substantially, becoming comparable to ProtoNets in some settings. With greater levels of noise and occlusion in the instance test, Pretrain+Tune+Replay is often superior to Protonets, demonstrating that these conditions evaluate different capabilities.

### 5.1 Comparison of Pretrain+Tune and Protonets methods

The experiments involved two base methods: Pretrain+Tune and ProtoNets (Snell et al., 2017). It's natural to compare their performance, but comparison should be cautious as ProtoNets and Pretrain+Tune methods do not perform the same learning task.

In ProtoNets, classification is achieved by comparing embeddings, which necessitates a short-term memory (STM) of the reference embedding being matched. By convention, that memory is in the testing framework rather than the ProtoNet architecture itself. Given this perspective, adding the STM (replay buffer) to the Pretrain+Tune network makes it conceptually more similar to ProtoNets, and the results are also more similar. In this study the STM is used for replay only. In future work, it could also be used for classification for recent samples still in short-term memory, as was done in AHA (Kowadlo et al., 2021).

Another way in which Pretrain+Tune and ProtoNet learning differs, is that ProtoNets do not actually acquire new knowledge during training and therefore do not continually learn. There is just one optimizer (for the meta-learning 'outer loop') that gets triggered during the 'pre-training' phase, meaning that it learns meta parameters for an embedding representation that will be optimal for downstream tasks.

### 5.2 Wide vs Deep – big batches vs many tasks (no replay)

The way that data are presented, not just the number classes, makes a difference to learning. When the number of classes was held constant at 20, but the number and size of support sets was varied (Baseline 2 vs Wide 1 or Deep 1), performance was better for the wide configuration. However, when the number of

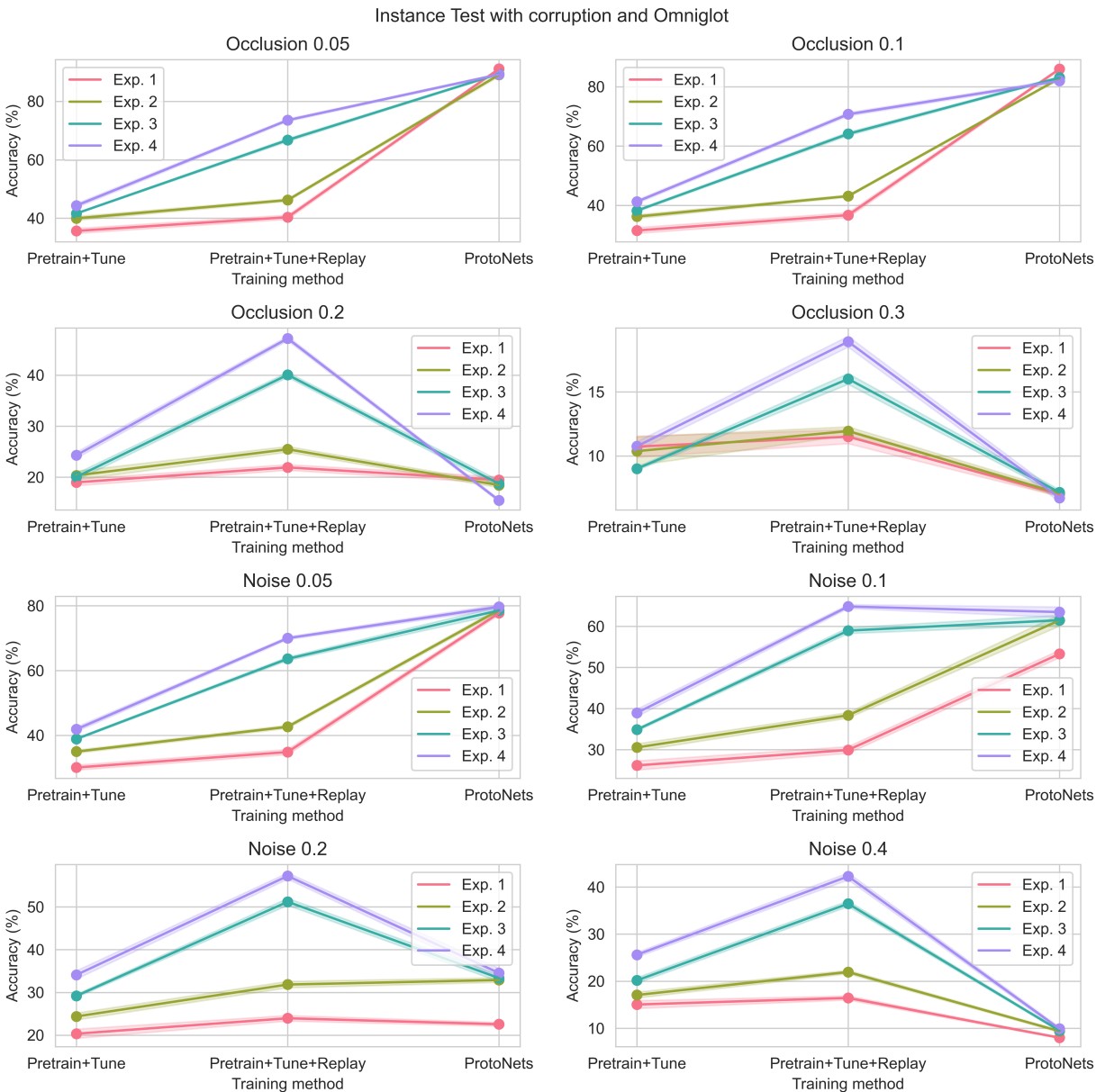

Figure 8: **Instance test: Omniglot images, with noise or occlusion.**

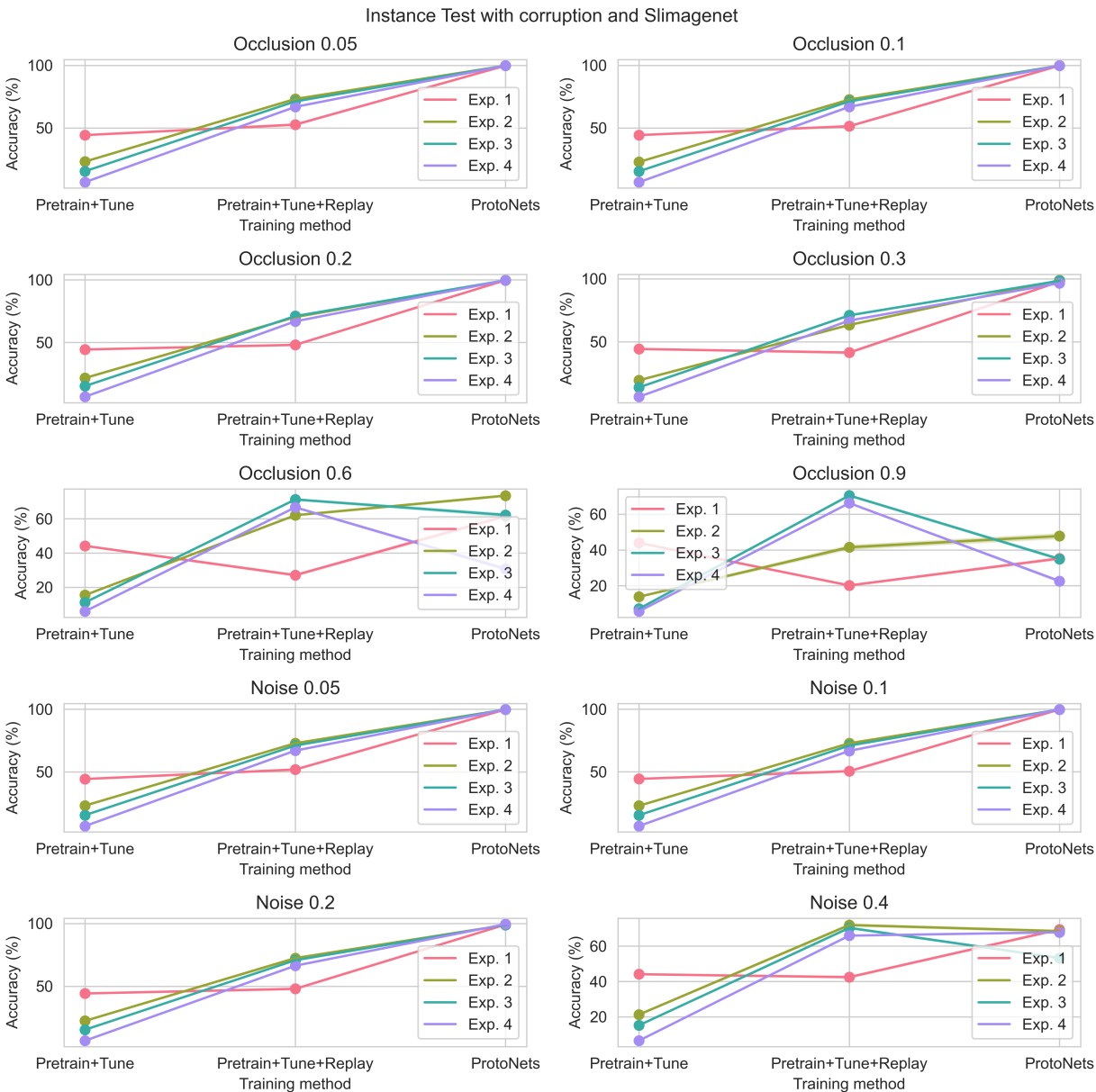

Figure 9: **Instance test: SlimageNet64 images, with noise or occlusion.**

classes was increased by an order of magnitude to 200 (Wide 2 vs Deep 2), performance was better in the deep configuration, where the classes were spread out across smaller batches. This was unexpected given that weight updates (in Pretrain+Tune) occur after each support set, and the more support sets there are, the more it could 'forget' earlier learning. It is possible that as the number of classes increase, the larger support sets (in the Wide experiments) are harder to learn, or cause sharper forgetting by virtue of the fact that more knowledge is being acquired in one update.

ProtoNets (Snell et al., 2017) are very effective in the scaling test, despite not actually learning during these tasks. This implies that an effective embedding space was learned during pre-training for the task. Since no learning takes place, performance cannot suffer due to forgetting. Therefore lower performance when there are a lot of classes, as in Wide 2 and Deep 2, is likely due to generation of similar embeddings for different classes.

## 5.3   Specific instances (no replay)

The results suggest that one-shot distinguishing of specific (very similar) instances is not more difficult than classification, for these methods and architectures.

The fact that Pretrain+Tune accuracy increases as the instances are distributed over more support sets, further hints that CNNs may be more effective at continual learning with smaller support sets, which is in-line with our interpretation of why Deep (more, smaller support sets) was easier than Wide in the scaling experiments (explained in more detail in Section 5.2 above).

ProtoNets are effective in the instance test as well as classification. In the instance test, generalization is not required, and so representations are less likely to overlap reducing the possibility of clashes. In addition, no fine-tuning occurs (see earlier in the Discussion, Section 5.1), so performance is very stable across all configurations. However, with the addition of noise or occlusion, this inflexibility reduced ProtoNet recognition performance significantly, especially for the Omniglot dataset. Protonets was less disrupted by noise and occlusion in the SlimageNet64 dataset, probably because these images have background content which provides stable features that allow recognition. The desirability of using background features for recognition depends on the intent of the task. In some cases, recognising the scene where an object was placed might be the only way to distinguish it.

## 5.4   Effect of replay

As hypothesized, adding replay to Pretrain+Tune enabled a strong improvement across tasks. Despite the improvement, performance did not reach the same level as ProtoNets in most scaling tests (classification). Replay had a more substantial impact in the instance test. In instance test conditions with higher levels of noise or occlusion, where the ProtoNets method was less robust, the Pretrain+Tune+Replay method often outperformed ProtoNets.

Although Pretrain+Tune+Replay did not convincingly outperform ProtoNets in many experiments, there are implications for future work. Replay does improve the performance of a statistical learner (i.e. an LTM) in few-shot continual learning.

Finally, and perhaps most significantly, ProtoNets as implemented do not acquire new knowledge during training, as explained earlier in this section, and therefore do not actually demonstrate continual learning. The inability to adapt is very likely to limit performance if there is a shift in the statistics of data distribution, or out of domain (e.g. as in OSAKA experiments (Caccia et al., 2020)). Pushing these limits and exploring weight adaptation during training in the context of CFSL is an important area for future work.

## 5.5   Limitations and future work

The base CFSL framework that we used, measures performance after all the learning has occurred. In contrast, most studies in the continual learning literature document progressive performance as new tasks are introduced, and these task changes might not be observable (e.g. Harrison et al. (2020)).

The instance test described and evaluated in this paper is limited by the sophistication and realism of the images used, and the limited challenge of synthetic noise and occlusion. In SlimageNet64 experiments, the background also provides a strong cue as to the correct figure match. More realistic high resolution video imagery would probably better capture the conflicting challenges of balancing memorisation-recognition and class-generalisation.

Despite these limitations, we observed that under greater noise and occlusion conditions, neither ProtoNets nor Pretrained methods with or without Replay were satisfactory in the Instance test. It is possible that use of more sophisticated architectures could help, comprising a promising direction for future work. Firstly, more recent architectures could be used for the feature extraction, such as ResNet He et al. (2016) and EfficientNet variants Tan & Le (2019). Secondly, more implementing more advanced replay mechanisms as described in related work (Section 2.4), such as selective storing and retrieval of memories into the buffer, or implementing the CLS (McClelland et al., 1995; O'Reilly et al., 2014; Kumaran et al., 2016) concept of dual pathways for pattern separation and generalisation as in AHA (Kowadlo et al., 2019; 2021). Finally, the success of Protonet training implies that inventing Protonet+Replay would be a promising direction.

# 6 Conclusion

Continual few-shot learning of both classes and instances is a necessary capability for agents operating in unfamiliar and changing environments. This study is one of the first steps in that direction, combining continual and few-shot learning, additionally evaluating the ability to recognise specific instances, and in doing so demonstrating that replay is a competitive approach under certain conditions.

We aimed to enhance the CFSL framework and evaluate a set of common CFSL approaches on the resulting tasks. The CFSL framework was scaled to 200 classes, to make it more comparable to typical continual learning experiments. We introduced two variants, **Wide** with fewer larger training 'support sets' and **Deep** with a greater number of smaller support sets. We also expanded the CFSL framework with a few-shot continual instance-recognition test, which measures a capability important in everyday life, but often neglected in Machine Learning.

We found that increasing the number of classes decreased classification performance (scaling test) and the way that the data were presented did affect accuracy. Performance in the few-shot instance test was comparable to few-shot classification, but results were significantly worse with the addition of basic challenges such as noise and occlusion.

Augmenting models with a replay mechanism improved performance substantially in most experiments. ProtoNet training was superior to pretraining with fine-tuning under most settings, with the exception of the instance test given high levels of noise or occlusion. This demonstrates that the instance test requires model qualities that are not evaluated under existing CFSL experimental regimes, and that in some of these conditions an LTM plus replay architecture may be preferable.

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

# 7 Appendix

## 7.1 Pretrain+Tune method

Unlike the standard configurations of VGG, there are a variable number of blocks, determined by hyperparameters. Additionally, the number of filters increase in each subsequent block similar to the VGG Simonyan & Zisserman (2014) architecture. However, they increase linearly instead of exponentially i.e. (32-64-96) rather than (32-64-128). Hyperparameters control the number of blocks and filters as well as the initial learning rate.

## 7.2 Fine-tuning steps

Adding the replay buffer increased the memory requirements. For some experiments, we reduced the number of fine-tuning training steps to make it possible to run within our hardware constraints. The number of steps are documented in Tables 5 and 6.

Table 5: **Fine-tuning for scaling test.** The number of fine-tuning training steps for the Pretrain+Tune+Replay scaling test.

| Experiment | Fine-tuning training steps |
| --- | --- |
| Baseline 1 | 120 |
| Baseline 2 | 60 |
| Wide 1 | 30 |
| Wide 2 | 30 |
| Deep 1 | 5 |
| Deep 2 | 5 |

Table 6: **Fine-tuning for the instance test.** The number of fine-tuning training steps for the Pretrain+Tune+Replay instance test.

| Experiment | Fine-tuning training steps |
| --- | --- |
| Exp. 1 | 120 |
| Exp. 2 | 120 |
| Exp. 3 | 60 |
| Exp. 4 | 30 |

## 7.3 Tabular results

## 7.4 Optimized hyperparameters

To facilitate fairer comparison of architectures, hyperparameter optimization was used to find the best configuration under each experimental condition. The resulting hyperparameter values used in experiments are recorded here.

Table 7: **Scaling test accuracy - Omniglot.** These results are for the best configurations found through hyperparameter search. Accuracy is shown in %, as mean ± standard deviation across 5 random seeds. NC=number of classes.

| Method name | Baseline 1 NSS=4, CCI=2, $n$-way=5, NC=10 | Baseline 2 NSS=8, CCI=2, $n$-way=5, NC=20 | Wide 1 NSS=4, CCI=2, $n$-way=10, NC=20 | Wide 2 NSS=4, CCI=2, $n$-way=100, NC=200 | Deep 1 NSS=20, CCI=2, $n$-way=2, NC=20 | Deep 2 NSS=80, CCI=2, $n$-way=5, NC=200 |
|---|---|---|---|---|---|---|
| Pretrain+Tune | $64.95 \pm 1.00$ | $33.71 \pm 3.54$ | $54.25 \pm 0.59$ | $4.22 \pm 0.50$ | $33.44 \pm 1.18$ | $6.64 \pm 0.68$ |
| Pretrain+Tune+ Replay | $81.15 \pm 0.81$ | $73.5 \pm 0.43$ | $80.74 \pm 0.70$ | $31.78 \pm 0.65$ | $60.94 \pm 0.90$ | $18.62 \pm 0.71$ |
| ProtoNets | $86.67 \pm 1.35$ | $88.04 \pm 1.12$ | $86.92 \pm 0.42$ | $65.61 \pm 9.31$ | $88.56 \pm 0.61$ | $80.30 \pm 1.15$ |

Table 8: **Scaling test accuracy - SlimageNet64.** These results are for the best configurations found through hyperparameter search. Accuracy is shown in %, as mean ± standard deviation across 5 random seeds. NC=number of classes.

| Method name | Baseline 1 NSS=4, CCI=2, $n$-way=5, NC=10 | Baseline 2 NSS=8, CCI=2, $n$-way=5, NC=20 | Wide 1 NSS=4, CCI=2, $n$-way=10, NC=20 | Wide 2 NSS=4, CCI=2, $n$-way=100, NC=200 | Deep 1 NSS=20, CCI=2, $n$-way=2, NC=20 | Deep 2 NSS=80, CCI=2, $n$-way=5, NC=200 |
|---|---|---|---|---|---|---|
| Pretrain+Tune | $15.85 \pm 0.20$ | $7.80 \pm 0.10$ | $9.40 \pm 0.19$ | $4.83 \pm 0.05$ | $5.89 \pm 0.01$ | $2.66 \pm 0.02$ |
| Pretrain+Tune+ Replay | $13.59 \pm 0.14$ | $9.79 \pm 0.16$ | $10.45 \pm 0.18$ | $4.76 \pm 0.04$ | $9.35 \pm 0.08$ | $4.28 \pm 0.06$ |
| Protonets | $25.72 \pm 0.17$ | $18.24 \pm 0.15$ | $18.04 \pm 0.10$ | $11.62 \pm 0.04$ | $19.29 \pm 0.16$ | $12.21 \pm 0.10$ |

Table 9: **Scaling test, best hyperparameters - Omniglot.** These results show best architectures found for each experiment, selected through hyperparameter search. A block consists of a 2d convolutional layer, a batch norm layer and a max pooling layer. The number of filters is for the first block, and it increases linearly for each subsequent block. The learning rate is also shown, denoted with lr. Unless specified, lr=0.01. In the case of the replay buffer, $b$ denotes the size of the buffer in support sets, and $p$ denotes the number of samples taken from the buffer for each fine-tuning support set.

| Method name | Baseline 1 NSS=4, CCI=2, $n$-way=5, NC=10 | Baseline 2 NSS=8, CCI=2, $n$-way=5, NC=20 | Wide 1 NSS=4, CCI=2, $n$-way=10, NC=20 | Wide 2 NSS=4, CCI=2, $n$-way=100, NC=200 | Deep 1 NSS=20, CCI=2, $n$-way=2, NC=20 | Deep 2 NSS=80, CCI=2, $n$-way=5, NC=200 |
|---|---|---|---|---|---|---|
| Pretrain+Tune | 128 filters, 3 blocks | 512 filters, 3 blocks | 256 filters, 3 blocks | 128 filters, 2 blocks | 256 filters, 3 blocks | 256 filters, 3 blocks |
| Pretrain+Tune+ Replay | 512 filters, 3 blocks, $b$=2, $p$=10 | 128 filters, 3 blocks, $b$=4, $p$=10 | 256 filters, 3 blocks, $b$=2, $p$=20 | 128 filters, 2 blocks, $b$=2, $p$=50 | 256 filters, 3 blocks, $b$=5, $p$=10 | 256 filters, 3 blocks, $b$=5, $p$=10 |
| ProtoNets | 128 filters, 4 blocks | 128 filters, 4 blocks, lr=0.001 | 128 filters, 4 blocks, lr=0.001 | 128 filters, 4 blocks | 128 filters, 4 blocks | 256 filters, 4 blocks, lr=0.001 |

Table 10: **Scaling test, best hyperparameters - SlimageNet64.** These results show best architectures found for each experiment, selected through hyperparameter search. A block consists of a 2d convolutional layer, a batch norm layer and a max pooling layer. The number of filters is for the first block, and it increases linearly for each subsequent block. The learning rate is also shown, denoted with lr. Unless specified, lr=0.01. In the case of the replay buffer, $b$ denotes the size of the buffer in support sets, and $p$ denotes the number of samples taken from the buffer for each fine-tuning support set.

| Method name | **Baseline 1** NSS=4, CCI=2, $n$-way=5, NC=10 | **Baseline 2** NSS=8, CCI=2, $n$-way=5, NC=20 | **Wide 1** NSS=4, CCI=2, $n$-way=10, NC=20 | **Wide 2** NSS=4, CCI=2, $n$-way=100, NC=200 | **Deep 1** NSS=20, CCI=2, $n$-way=2, NC=20 | **Deep 2** NSS=80, CCI=2, $n$-way=5, NC=200 |
|---|---|---|---|---|---|---|
| Pretrain+Tune | 64 filters, 4 blocks | 64 filters, 4 blocks | 64 filters, 4 blocks | 128 filters, 4 blocks | 64 filters, 4 blocks | 64 filters, 4 blocks |
| Pretrain+Tune+ Replay | 256 filters, 4 blocks, $b$=2, $p$=20 | 64 filters, 4 blocks, $b$=2, $p$=20 | 64 filters, 4 blocks, $b$=8, $p$=10 | 128 filters, 4 blocks, $b$=8, $p$=12 | 64 filters, 4 blocks, $b$=2, $p$=5 | 64 filters, 4 blocks, $b$=4, $p$=10 |
| ProtoNets | 64 filters, 4 blocks | 64 filters, 4 blocks | 64 filters, 4 blocks | 64 filters, 4 blocks | 128 filters, 4 blocks | 64 filters, 4 blocks |

