# OpenReview forum: "Expanding continual few-shot learning benchmarks to include recognition of specific instances"
_TMLR — Rejected by TMLR_

### Review · Reviewer_D1tq · 2023-08-06

**Summary Of Contributions:**

In this study, the authors delve into an emerging research area that combines continual learning with few-shot learning (CFSL). This fusion enables the potential addressal of challenges related to agent operations. The authors improve evaluation for the CFSL framework by expanding the class number and introducing two novel dimensions: 'Wide' and 'Deep'. In addition, they identify limitations in current research, notably CFSL methods are vulnerable to noise and occlusion.

**Audience:**

No

**Claims And Evidence:**

Yes

**Requested Changes:**

1. Descriptions for several figures, including Figure 8 and 9, are absent. Given the new research field, it's crucial to provide more setting and description for each experiment.

2. While the paper is overall well-presented, many figures and tables occupy excessive space. I recommend that the authors streamline the paper by relocating less crucial content to the supplementary section. This will enable readers to focus on the essential parts and enhance clarity.


**Strengths And Weaknesses:**

1. The paper is well-organzied and give very details about backgroup knowledge
2. The authors coduct a wide range of experiments, including an analysis of noise and occlusion.

Weakness:
1. While the authors provide examples to elucidate the concept of CFSL, these examples come across as abstract and weak. Could the authors offer more tangible real-world scenarios or applications to show the importance of CFSL research?

2. From my perspective, even with an increase to 200 classes, it remains insufficient to adequately demonstrate the strengths of continual learning and few-shot learning. I'd suggest, if possible, beginning with a class size of 1k and scaling it up to roughly 10k.

---

> ### Author Response · Authors · 2023-09-19
> **Response to review**
>
> Thank you for taking the time to review our paper and for the constructive feedback. Thank you for identifying the strengths of a well organized, detailed paper, and thorough experiments.
>
> Regarding the requested changes, we will add descriptions for the captions in the figures. Thanks for the suggestion of streamlining the paper, we can move less central material to a supplementary section.
>
> The other aspect of *Requested Change 1* is related to Weaknesses 1.
>
> > Given the new research field, it's crucial to provide more setting and description for each experiment.
>
> and
>
> > While the authors provide examples to elucidate the concept of CFSL, these examples come across as abstract and weak. Could the authors offer more tangible real-world scenarios or applications to show the importance of CFSL research?
>
> Thanks for the suggestion to offer a more compelling explanation of the importance of CFSL. Additional examples include the following. In contrast to most ML models, animals are able to learn classes and instances quickly, which is an important skill for agents that are to succeed in realistic tasks. For example, a pick-and-place robot should be able to recognize and handle new products after being shown just one demonstration. Moreover, reasoning about specific instances, as in the Instance Test, is also important. For example, we may require the pick-and-place robot to put all the cereal boxes in a specific bin.
>
> Regarding Weakness 2:
>
> > From my perspective, even with an increase to 200 classes, it remains insufficient to adequately demonstrate the strengths of continual learning and few-shot learning. I'd suggest, if possible, beginning with a class size of 1k and scaling it up to roughly 10k.
>
> Although this would be worthwhile, in this study we aimed to make early CFSL experiments compatible with a scale commonly accepted in the field of Continual Learning and to broaden experimental scenarios to the Instance Test. We believe that the set of experiments was adequate to learn about strengths and weaknesses of a range of methods and it is therefore worthwhile to share the results with the community. We agree that increasing the number of classes from 1k to 10k could be a good direction for Future Work, which we will add to the paper.

---

### Review · Reviewer_MHrv · 2023-09-05

**Summary Of Contributions:**

The manuscript provides an in-depth investigation on continual few-shot learning, a continual learning task where the number of examples for each class is low. The authors considers a few variations in the task set-up, including training on a stream of fewer class sets where each set has many classes (wide) vs. training on a stream of more class sets where each set only has a few class. The experiments evaluates two approaches (proto-nets vs pretrain+finetune) on these set-ups, and investigates the effects of replay.

**Audience:**

Yes

**Claims And Evidence:**

Yes

**Requested Changes:**

Please provide more explanation of the "instance test" setting.

**Strengths And Weaknesses:**

Strength:
+ Clear discussion of the task and different set-ups (except for a few places mentioned below)
+ Thorough experiments and analyses

Weakness:
+ The "instance test" is a bit confusing. As it is described, is it simply trying to find exact duplicates? If so, then it seems that this is a quite contrived setting, as this is more like a compression & retrieval task rather than a learning task.

Disclaimer: I do not work in continual learning. While I tried to provide a best review with my knowledge, there may be something that I am missing (e.g., important related works / methods).

---

> ### Author Response · Authors · 2023-09-19
> **Response to review**
>
> Thank you for taking the time to review our paper and for the constructive feedback. Thank you for identifying the strengths of a clear discussion and thorough experiments.
>
> Regarding the instance test, we will review and improve the description and provide more examples, which will address the identified ‘Weakness’ and *Requested Change*.
>
> We can emphasize that it is not simply trying to find duplicates, since varying levels of random noise and occlusion are applied (different patterns between train and test). It is a setting that has largely not been investigated in ML, and yet it represents many important real world situations. For example, a person can easily identify their own specific cup of coffee, or their own locker, and not see them simply as a generic cup or locker class. For a tangible robotics example, take a pick-and-place robot that should place items in a specific bin (rather than any generic bin). Although it may seem easy, the experiments showed that introducing noise and occlusion was deleterious, affecting the methods differently. Therefore, we thought it worthwhile to include in the study.

---

### Review · Reviewer_aKoc · 2023-09-11

**Summary Of Contributions:**

This paper addresses the continual few-shot learning problem, which is of great importance to the machine learning field. The authors modify CFSL to make it more comparable to standard continual learning protocols and introduce the recognition of specific instances of classes in CFSL. The main contribution of this paper is vague, and this paper looks more like an empirical study of different CFSL methods.

**Audience:**

Yes

**Broader Impact Concerns:**

No Broader Impact Concerns.

**Claims And Evidence:**

Yes

**Requested Changes:**

This paper requires at least a major revision. According to my previous comments, some concerns are needed to be addressed:
1. The setting should be highlighted with mathematical notations and explanations to give the audience a holistic overview of the background.
2. Making clarifications about the differences between CFSL and FSCIL and making these settings compatible.
3. To strengthen the contribution (which I think cannot be achieved by a major revision). I can’t entirely agree with the authors on the contribution of this paper, and perhaps more baselines, state-of-the-art methods, and empirical evaluations are needed.

**Remark**
1. The authors contain a caption of Figure 1 in the figure, e.g., “Visual representations … set labels.” These sentences should emerge in the caption instead of the image.


[1] Few-shot class-incremental learning. CVPR 2020
[2] Forward compatible few-shot class-incremental learning. CVPR 2022
[3] Few-shot class-incremental learning by sampling multi-phase tasks. TPAMI 2022
[4] Few-shot incremental learning with continually evolved classifiers. CVPR 2021
[5] Subspace regularizers for few-shot class incremental learning. ICLR 2022

**Strengths And Weaknesses:**

Pros.
1. Continual few-shot learning is essential to the machine learning field.
2. The authors enhance CFSL and make it compatible with continual learning benchmarks.
3. This paper evaluates the ability of typical CFSL methods with typical continual learning benchmarks.

Cons.
1. Unfortunately, the problem of continual few-shot learning is not even properly defined in the paper. For example, Section 3.1 introduces continual few-shot learning while still confusing. I may assume that the audience has no background about the CFSL setting, and a proper definition is essentially needed for the current setting.
2. I am unsure about the difference between the current continual few-shot learning problem and the few-shot class-incremental learning problem (FSCIL) [1-5]. It seems both of them address the learning process of new (few-shot) classes without forgetting the former ones. Hence, a detailed discussion is needed in the related work for clarification.
3. Actually, I find the contribution quite limited. This paper looks more like an empirical study of several typical methods in CFSL, while only three methods are reproduced in the current setting. If the authors claim the contribution lies in the open-source code, the code is also not attached (even in the anonymous mode), making it hard to evaluate the contribution. Besides, I am also curious that the authors only reproduce the typical meta-learning baselines in the current setting, without any current state-of-the-art.
4. Using memory to overcome forgetting is somehow common sense in the continual learning field, and these experimental results are not surprising but quite normal and lead to no new insights.

---

> ### Author Response · Authors · 2023-09-18
> **Response to review**
>
> Thank you for taking the time to review our paper and for the constructive feedback. Thank you for finding the pros, tackling an important field and making these CFSL experiments compatible with the field. We address the cons below.
>
> **CFSL Definition**
>
> Cons 1 and 2 relate to defining CFSL and clarifying the difference with FSCIL. Thanks for the comments. We will expand the related work to define Continual Learning more broadly for those readers that are not familiar. Included in that will be mathematical notation, and explanation for how CFSL is distinct from FSCIL. They share a common objective, but training and test procedures are significantly different (Antoniou et al. 2020).
>
> Making those changes will address *Requested Change 1 and 2*.
>
> **Contribution**
>
> Con 3, 4 and *Requested Change 3* pertain to a limited contribution. We understand the sentiment. However, with respect, we believe that there is a contribution worth sharing with the community, see more detailed response below:
>
> > Actually, I find the contribution quite limited. This paper looks more like an empirical study of several typical methods in CFSL, while only three methods are reproduced in the current setting.
>
> The three methods are representative of different families of approaches. Also, the main contribution was to introduce a new benchmark, the instance test. As argued in the paper, it is an important yet neglected setting. We will improve the text to make this clearer.
>
> All together, including different levels of noise and occlusion, there are already a large number of experiments, and we believe they provide a solid foundation for an early study on CFSL. Future work by us and others could broaden the number of methods investigated, including newer state-of-the-art methods mentioned in the reviewer's next comment (below); as the state-of-the-art is rapidly advancing.
>
> > If the authors claim the contribution lies in the open-source code, the code is also not attached (even in the anonymous mode), making it hard to evaluate the contribution. Besides, I am also curious that the authors only reproduce the typical meta-learning baselines in the current setting, without any current state-of-the-art.
>
> The code is not the main contribution, as we only tweaked the open source code that is available from Antoniou et al. (2020). We will share the URL when it can be de-anonymised or share it in an anonymous fashion beforehand. Also, we will make it clear that the source code is not the main contribution.
>
>
> > Using memory to overcome forgetting is somehow common sense in the continual learning field, and these experimental results are not surprising but quite normal and lead to no new insights.
>
> Memory-replay is one of the main approaches to continual learning (Parisi et al. 2019). While we agree that adding memory-replay was likely to enhance learning in CFSL setting, we felt it worthwhile to confirm and quantify the degree of improvement achievable in a range of experimental conditions. While replay did help, it has limitations, and we feel it is worthwhile to share these learnings.
>
> **Remarks**
>
> Thanks for picking that up, we will make that change.

---

### Author Response · Authors · 2023-07-13
**Full response to reviewers**

I said above that I would add full responses to the reviewers in the previous submission. However, we had the same character limits. The full responses are available on request (if there is a way to submit them).

---

### Author Response · Authors · 2023-11-07
**Updated paper**

Thank you to the action editor and reviewers for their time and constructive feedback.
We made many of the recommended changes and updated the arXiv copy here: https://arxiv.org/abs/2209.07863

---

### Decision · Action_Editor_eXZy · 2023-10-18

**Recommendation:** Reject

**Comment:**

This paper is a TMLR re-submission (it previously went through a round of reviews and the authors withdrew it to have more time to make the changes). Since then, the authors have added a discussion of related literature, made some clarifications to the experimental setup and added more experiments (an additional dataset) which is a substantial improvement.
Some of the feedback of the reviewers of the previous round remained unaddressed, notably adding more baselines (the current draft still has only two baselines). This is also something that current reviewers identified (Reviewer aKoc).

The reviewers of the current submission found the topic interesting and recognize that the authors conducted a wide range of experiments (Reviewer D1tq and Reviewer MHrv) but they also identified a set of weaknesses: clarity of the problem definition can be improved, and its relationship with related tasks like FSCIL (Reviewer aKoc), the motivation of studying CFSL generally and the instance test in particular should be strengthened, by providing examples of real-world applications and additional baselines (Reviewer aKoc). Reviewer MHrv also raised the issue of the instance test being unclear. In their response to Reviewer MHrv, the authors say about the instance test that "We can emphasize that it is not simply trying to find duplicates, since varying levels of random noise and occlusion are applied". In this case, I would recommend adjusting Figure 3 accordingly, and emphasizing this in the revised text.

Overall, I agree with the reviewers that the clarity of the paper should be improved by adding a clear problem description and discussion on how it relates to other related problems (the authors may consider, for example, having a separate section in the paper to explain the problem setup, rather than including that in the experimental results section). It would also be really useful to discuss state-of-the-art methods for FSL and CFSL in the related work (the FSL section currently only discusses MAML), and include some of those methods to the empirical evaluation, as that would strengthen the claims made. So, while I think the paper is significantly improved from the previous iteration, it requires a major revision to take this feedback into account. I recommend that the authors consider making a major revision at a later time.

**Audience:**

Yes, the topic of this paper lies at the intersection of continual and few-shot learning which are important problems that several individuals in the community are interested in.

**Claims And Evidence:**

This paper proposes a new benchmark for evaluating Continual Few-Shot Learning (CFSL) methods by extending previous CFSL benchmarks in two ways: increasing the number of classes and adding an instance test (where the goal is to recognize specific instances that were previously seen rather than learning new classes).

This paper's main contribution is proposing these modifications to CFSL benchmarks and conducting an empirical study where they investigate how two methods (a pretrain+fine-tune approach and a Prototypical Network) behave under different settings within the CFSL framework. For example, they study how performance changes as the number of classes increases (in two configurations with which new classes are introduced across support sets; 'wide' vs 'deep'). They also study the performance on their proposed instance test (including the vulnerability to the addition of noise and occlusion in that context), and also the effect of adding replay (in the context of the pretrain+finetune approach only).

Because of the limited scope of the evaluation when it comes to the set of approaches considered (they consider only 2 approaches, both of which are far from the state-of-the-art), it is hard to really substantiate claims about the difficulty of the task as the number of classes changes, or the relative difficulty of the instance task versus the standard CFSL tasks, or the relative merit of models e.g. "replay is a competitive approach". It seems that, in order to be able to draw a set of generalizable findings from this investigation, it is necessary to add more baselines / methods. For instance, the authors discuss that there are major differences between the two methods they consider (making it hard to draw conclusions about which aspect of a given method is responsible for the observed performance characteristics), which indeed points to the fact that the addition of more methods would be very helpful. For example, MAML is a meta-learning method that does adapt the embedding in each episode, unlike Prototypical Networks; and considering more recent / state-of-the-art (C)FSL methods would be very helpful and informative too. This would allow to better substantiate all the above claims.

**Resubmission Of Major Revision:**

The authors may consider submitting a major revision at a later time.